# Male allyship to advance women's leadership in global health academia: A qualitative study

**Amanda Marr Chung**👤iD⊛*, **Ola Alani**iD⊛, **Michele Barry**

Stanford Center for Innovation in Global Health, Stanford University, Stanford, California, United States of America

⊛ These authors contributed equally to this work.
* amarr@stanford.edu

## Abstract

Women are underrepresented in leadership positions within global health. Although women leaders have been shown to foster inclusive work environments and prioritize improvements in women's health, they face barriers to their advancement, including microaggressions and disproportionate caregiving responsibilities. Male allyship can facilitate the elevation of women into global health leadership roles. This study explores the experiences of global health leaders in academia of male allyship and identifies actions and best practices to support the growth of women's leadership in global health. Qualitative semi-structured interviews were conducted with twenty-one global health leaders (11 females, 10 males) from U.S. and Canadian academic institutions. Interviews were recorded, transcribed, and coded using a combined inductive-deductive approach. Participants identified barriers and outlined potential approaches to support women's advancement to leadership roles. For the individual male ally, recommendations included completing a self-assessment (to mitigate counterproductive behaviors and biases), engaging in effective mentorship practices, advocating publicly, and serving as a positive role model. Recommendations at the institutional level emphasize the importance of cultivating an enabling environment that facilitates open dialogue, establishing goals and metrics; and implementing allyship training with periodic evaluation. At the societal level, participants suggested promoting early education and shared caregiving to shift cultural norms on gender roles. This paper provides a framework of actions and resources to cultivate and support male allyship for women's leadership advancement in global health. Effective male allyship begins with acknowledging power dynamics and an understanding of how intersectionality, beyond gender alone, shapes women's careers and workplace dynamics. Additionally, mentorship and collaborative peer support are critical to promoting women's career development. Individual allyship when combined with institutional and societal actions and policies, can facilitate the advancement of women in global health leadership roles.

**Data availability statement:** Data is available upon request from Stanford irbeducation@stanford.edu. Despite de-identifying the data, the authors determined it cannot be made publicly available for ethical reasons (e.g., public availability would compromise participant privacy) as data contain potentially identifying information related to participants' background, expertise, and occupation.

**Funding:** This work was supported by WomenLift Health/New Venture Fund (NVF-STA25-WOLH1-2023-11-01 to MB). The funders provided input into the interview participant list and interview guide but otherwise had no role in data collection and analysis, decision to publish, or preparation of the manuscript. AMC and OA received partial salary support from WomenLift Health/New Venture Fund to conduct this research.

**Competing interests:** OA has declared that no competing interests exist. MB and AMC have read the journal's policy and have the following competing interests: MB is the founder and Honorary Board Member of WomenLift Health's Global Advisory Board. AMC is a North American WomenLift Health Fellow. This does not alter our adherence to PLOS ONE policies on sharing data and materials. There are no patents, products in development or marketed products associated with this research to declare.

## Introduction

The world suffers from a severe shortage of women leaders: women make up close to 70% of the global health workforce but only hold 25% of senior leadership positions [1,2]. Women in the workplace experience macroaggressions in the form of sexual assault and exploitation, the gender pay gap, and lack of paid family leave, with the U.S. being one of few countries in the world that do not have national paid maternity leave [3–5]. Microaggressions in professional settings show up when a woman is questioned about her qualifications, not acknowledged for her contributions, and assumed that she is not the leader among a mixed gender group [6,7]. According to the annual Global Health 50/50 report, gender parity has improved since 2018. For example, women CEOs increased from nearly 29% to 35% from 2018 to 2024. However, the world is still far from having an equal number of women as CEOs, senior managers, and board chairs [8].

The political climate has a profound effect on the role of women, which was already fragile at best. For example, in the United States, the dearth of women in leadership positions will likely be exacerbated by the ending of diversity, equity, and inclusion programs in 2025 [9,10]. In 2023, women were removed from high profile university president roles as a reaction to student protests on college campuses [11]. The current U.S. administration will also threaten women's rights to bodily autonomy, building on the overturning of Roe vs. Wade, which protected women's rights to abortion for fifty years [12].

Underrepresentation of women leaders and women's lack of access to supportive networks and sponsors are examples of second-generation gender bias, defined as "powerful but subtle and often invisible barriers for women that arise from cultural assumptions and organizational structures, practices, and patterns of interaction that inadvertently benefit men while putting women at a disadvantage" [13]. In addition to these barriers, mid-career women, especially women of color, tend to be in the sandwich generation, where they have primary caregiving responsibilities for children and aging parents [14,15]. This additional role comes at a time when mid-career women typically need to lean in to get promoted or tenured [15,16]. There is a stark disparity between the U.S. and other high-income countries, which have an average of over 18 weeks of paid maternity leave [17]. In the U.S., maternity leave is recognized by workplaces, but it is not always paid [18,19]. Moreover, other conditions specific to women including premenstrual and menopausal symptoms, miscarriages, and abortions merit flexible and paid leave organizational policies, but few provide such accommodations [5,20]. Women pursuing careers in global health often face extended timelines to attain senior positions because these roles typically require fieldwork. This requirement can coincide with the period when many women seek to start families or already have young children [21]. Consequently, these barriers contribute to a phenomenon known as the "leaky pipeline", in which women are more likely than men to leave academia before securing tenure [22].

### Benefits of women leaders

"Diversity drives innovation – when we limit who can contribute, we in turn limit what problems we can solve." -Telle Whitney

Historically, women have been excluded from participating in various roles in society including academia, research, and politics [23–25]. Women leaders create more equitable, inclusive, collaborative workplaces and bring different perspectives and approaches than men, including increased empathy and ethical initiatives [26–28]. These approaches have supported climate mitigation and been linked to lower $CO_2$ emissions [29]. Women leaders prioritize the needs of girls and women, vulnerable populations who are often overlooked in policies and research and are more at-risk for diseases such as HIV and malaria [21,30,31]. In research, women principal investigators were shown to increase women participants [32]. In the private sector, gender and racial diversity make good business sense, impacting the bottom line and increasing innovation, sales, revenue, and customers [26,33]. Caregiving responsibilities can make women better leaders, rather than parenthood being perceived as a liability [34–36]. A survey conducted by the Rutgers Center for Women in Business found that respondents doing unpaid caregiving developed and improved workplace skills such as empathy, efficiency, and task prioritization [35].

## Expanding the pool of male allies

Allyship is a process and journey requiring humility whereby individuals with more privilege and power center the needs of beneficiaries, respect them as equals, and act to create a more inclusive, engaging environment for those who have less access to resources and social power [37,38]. As a dominant group, men have a responsibility to use their power and privilege to sponsor women and elevate them to leadership roles. However, they must not see all women as a "monolithic oppressed group with white straight women as prototypical in our ethnocentric and heterocentric world" [39]. Similarly, the privilege men possess will vary, as some men may not be considered part of a dominant group due to other identities. According to Crenshaw's theoretical framework of intersectionality, identities such as "race, gender, [ability], and sexual orientation work together to shape people's lived experiences" [40]. Such overlapping identities may also influence whether males choose to act as allies.

Regardless of individual motives, society suffers from a lack of prominent male allies, often drawing on the same few men to speak out and serve as role models [41]. Moreover, men perceive themselves to be better allies than they actually are [42]. For example, in the 2022 Allyship-in-Action Benchmark Study, 71% of men reported that men "always" or "frequently" give credit to women for their ideas and contributions while only 40% of women agreed with the statement [42]. Men can also support women by stepping up their caregiving, whether it is for children or the elderly.

The purpose of this research was to gather concrete actions at the individual, organizational, and societal levels that male allies can take to become advocates of women as leaders. These recommendations can serve as a toolkit for men, with an interim goal of expanding the pool of male allies for women leaders in global health and ultimately achieving greater gender parity in global health leadership.

## Methods

### Study design

The research team compiled a list of potential participants based on the PI and Co-PIs' connections and knowledge of leaders in global health and a WomenLift Health directory of mid- and senior career academic global health leaders. The list was expanded through snowball sampling and online directory searches of academic and clinical institutions with a global health focus. The team utilized semi-structured interviews and developed two interview guides in English, one for each gender (male and female). The guide included the purpose of the study; adapted definitions of male allyship, mentorship, and sponsorship (Table 1); demographic questions; and interview questions [interview guide in S1 Text].

### Data collection

The study team employed purposive sampling and recruited male and female leaders in global health working in a United States or Canada-based academic or clinical institutions between January and November 2024. The team considered

PLOS Global Public Health

**Table 1. Terms and definitions.**

| Term | Definition |
|---|---|
| Male allyship | Purposeful collaboration of dominant group having power and privilege, men, with subordinate group having less power and privilege, women, to achieve gender equity by promoting women's voices and accomplishments [42]. |
| Mentorship | A professional relationship where an individual mentor with longer or more specialized work experience provides guidance, knowledge, expertise, and support to help a junior colleague develop their skills, navigate their career and improve on personal growth. |
| Sponsorship | A specific action in which a more senior person facilitates opportunities for an individual to gain visibility that might lead to career benefits, including but not limited to accomplishments required for academic promotion (adapted) [43] |

the career stage/years of experience, area of study, gender, and ethnicity of participants to collect diverse perspectives. The study team approached participants via email and invited them to participate. The interview guide and written consent forms were shared with the participants prior to the interview. All participants provided their written consent via email, and their verbal consent was recorded. Interviews were virtually conducted in English by the two co-PIs (AMC and OA) via Zoom and captured on an external recorder. On average, interviews lasted 50 minutes. The study team concluded interviews when participants' experiences and recommendations repeated across multiple interviews, thereby achieving data saturation.

## Data analysis

All interview audio recordings were transcribed using Rev AI transcription, and OA reviewed and edited for accuracy. Any identifying information was redacted prior to coding. Data were analyzed by AMC and OA, who have been formally trained in qualitative research. A combination of inductive and deductive coding along with axial and open coding were used. A code book with definitions was created using an individual and collaborative approach. Researchers created an analysis table in Excel utilizing codes to create themes and sub-themes with illustrative quotes.

## Ethical statement

This study was approved by the Stanford University Institutional Review Board, (#72715). All participants received a copy of the consent form via email prior to the interview date, and 100% gave verbal consent before starting the interview.

## Results

Forty-seven people were approached: twenty-one agreed to participate (45%); twenty-five did not respond; and one person declined to participate. There was near even representation of males and females, with a majority of participants identifying as white (Table 2). Participants are referred to by coded identifier, gender identity, and five-year age range (e.g., P1, F, 40–44) to preserve anonymity.

Participants spoke about challenges that prevent women from advancing to leadership roles in global health and shared recommendations to mitigate some of the barriers. These recommendations were grouped into themes (Fig 1). The study team identified several overlapping themes within the barriers and facilitators and aggregated them based on type of approach and behavior, and whether they were on an individual, institutional, or societal level. Some participants recommended resources and research such as papers, books, organizations, and checklists. The authors have referenced these where appropriate. With the exception of biological differences between the sexes, such as understanding the impact of menstruation and menopause on women and having policies to accommodate them, there were no differences in recommendations between males and females.

**Table 2. Demographics.**

| | n (%)* N=21 |
|---|---|
| **Gender** | |
| Male | 10 (48%) |
| Female | 11 (52%) |
| | |
| **Ethnicity** | |
| White[a] | 13 (62%) |
| Black[b] | 3 (14%) |
| South Asian (Indian)[c] | 4 (19%) |
| Indigenous | 1 (5%) |
| | |
| **Age Range** | |
| 40-44 | 2 (10%) |
| 45-49 | 4 (19%) |
| 50-54 | 4 (19%) |
| 56-59 | 3 (14%) |
| 60-64 | 2 (10%) |
| 65-69 | 5 (24%) |
| 70-74 | 1 (5%) |
| | |
| **Institutional Location** | |
| United States | 18 (86%) |
| Canada | 3 (14%) |

*Percentages are rounded to whole numbers

[a]Includes respondents identifying as white-Jewish, white-British, and Caucasian/white

[b]Includes respondents identifying as African and African American

[c]Includes respondents identifying as Indian American

## Allyship

Although the research team defined allyship in the methods, this definition needed further refinement, which they gathered from several interviews. This additional detail included expectations, actions, and behaviors at the individual and organizational levels. Allyship is earned and not something one can self-identify, according to one respondent with over 25 years of experience in global health research and education.

   "You should never call yourself an ally. You need to earn that someday." -P2, M, 50–54

Individuals who strive to be allies need to do a personal assessment, possess self-reflexivity and intentionality, commit to equity, and recognize and give up power, privilege, and implicit biases. At the organizational level, allyship needs an enabling environment, including promotion by leadership and a culture of embracing diversity.

## Intersectionality

A recurring theme from the interviews was the relationship between intersectionality and allyship. White cisgender, straight men benefit from privilege that also extends but less so to white, cisgender, straight women. When the

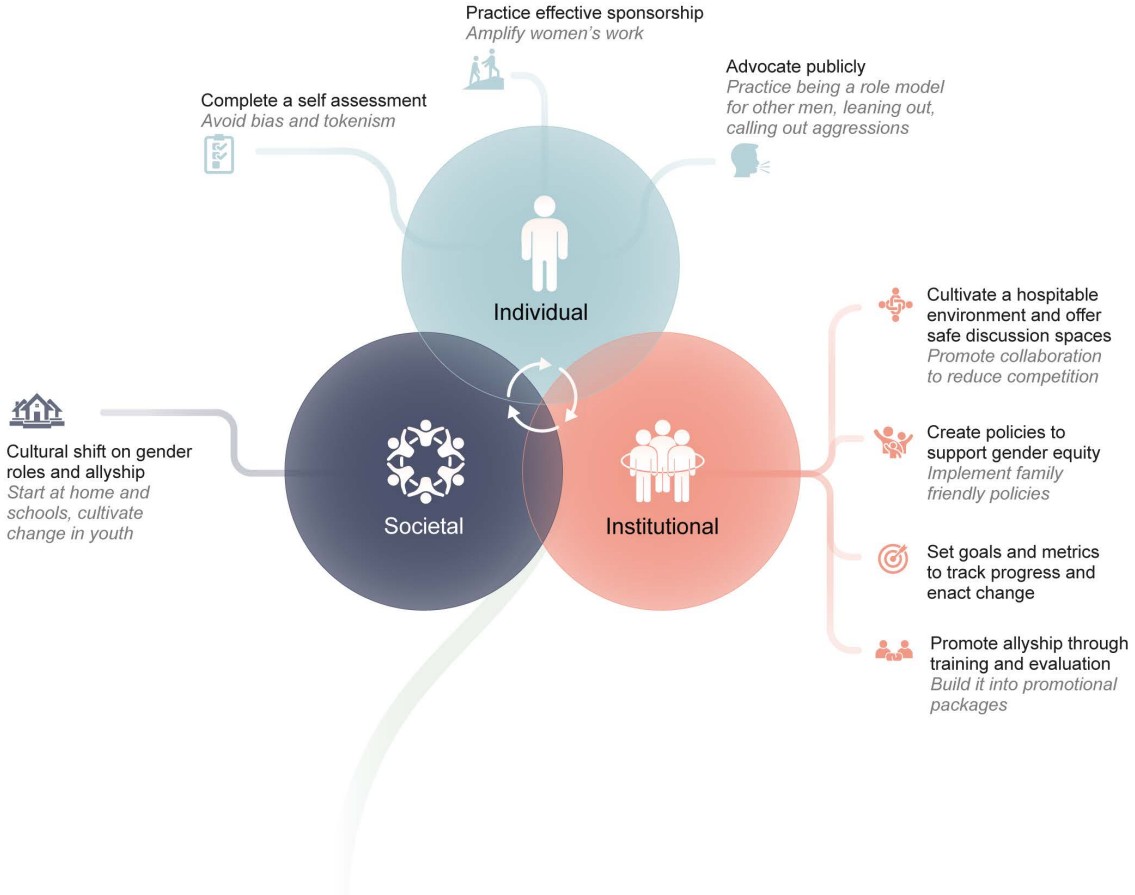

**Fig 1. Overview of recommendations for male allyship to promote women's leadership.**

respondent identified as part of a racial/sexual minority group, they had more awareness of the challenges women in leadership faced and increased solidarity with them. Indigenous mentees in particular felt imposter syndrome. Furthermore, the Me Too movement is a consideration for some but not for all males who identify as straight. For those who are concerned about the misinterpretation of their behavior, they take precautions in their communications and actions, selecting public meeting places and avoiding physical contact and closed door or weekend in-person meetings. Additionally, intersectionality plays a role in finding the right mentor-mentee relationship and is not gender-based.

## Motivation and incentives

Respondents identified several strategies to motivate men to become allies. Intrinsic motivations include being a father to a daughter, personal experience, and the desire to maintain a positive self-image. Additionally, extrinsic incentives, such as recognition and rewards, can encourage men to engage in allyship. A male participant with 45 years of experience in global health research, education, and clinical care discussed motivation for men to be better allies:

"Men who care about what their environment is like at work, they care about the way they're perceived and received. And so that's incentive enough." P1, M, 70–74

## Individual actions

**Self-assessments and avoiding counterproductive actions.** According to respondents, individual actions start with self-assessments related to the power and privilege that males have to relinquish. Once males accept and recognize this, it is their responsibility to lean out, create space, and sponsor underrepresented groups, especially women of color from the Global South.

In order to avoid counterproductive and ineffective efforts, males can:

- Discourage women from taking glass cliff positions (promotions and high-risk leadership positions that are likely to fail [44]).

- Highlight women's technical expertise in a public facing role.

- Give leadership opportunities with decision-making power to women, providing guidance and support.

Related to the last point, one respondent noted:
"…women in those roles end up taking on more and more roles and actually sometimes to the detriment of their career. So it seemed to be a leadership opportunity, but honestly, it's a service opportunity… I think the flip side of that is that also takes time away from one's ability to focus on the things you have to do to get promoted...it looks good because women are in those leadership roles, but actually if you kind of dig beneath the surface...There's nothing that comes off their plate in order to play those roles, and that's a problem." -P6, F, 50–54

## Mentoring and sponsorship

Participants also expanded on the definition of mentorship that the study team provided. Some respondents emphasized the importance of having personal relationships with their mentors/mentees and being a role model. A checklist was compiled from suggestions by the respondents (Table 3), including these statements:
"I want my mentees to feel comfortable asking things of me, and I want to feel comfortable asking things of my mentees. And even more importantly, we've got to feel comfortable with reciprocal constructive criticism. Otherwise, you're not a mentor, you're just a supporter" -P3, M, 60–64
"Women are just so much less likely to promote themselves... when you read an application from a woman, they'll underplay everything they've done. And when you read an application from most, many men, not all men, they exaggerate what they've done enormously." -P4, F, 65–69
"There's a wonderful book... It's about how men and women talk and how they hear language. And I think better understanding on both sides about those differences in communication styles would go a long way...."-P14, F, 65–69

## Meetings and panels

Respondents agreed that panels should be intentionally designed with gender equity in mind. They recommended that organizers, speakers and panelists, specifically men, should review meeting and conference panel invitation lists to ensure gender representation. Overwhelmingly participants thought men in leadership need to be vocal about not participating in manels (an all-male panel). A couple respondents referenced public statements by prominent global health leaders pledging not to be on a manel [49]. However, one participant spoke about understanding the audience and context before completely dismissing being on a manel, e.g., the only opportunity to access a certain audience. Other participants highlighted the importance of bringing on qualified speakers to avoid tokenism, as illustrated by this quote:
"So I think for example, when you're putting together a program…a seminar…a [disease X] conference, it's important to pay attention to gender equity, but it's really important to…make sure that …there are leading women scientists …and not as an afterthought...So I think tokenism is a challenge and it's important to really avoid that." -P8, M, 65–69

**Table 3. Checklist of actions that make good mentors for women.**

| Action | Description |
|---|---|
| Sponsor and mentor | Identify opportunities for visible leadership roles and provide the necessary guidance for success, including clear communication on time commitment and expectations for taking on the role |
| Recognize gender dynamics | Explain power structures and gender dynamics in organizational and network decision-making |
| Be aware of academic housework | Understand what academic housework is (undervalued service work such as mentoring and administrative tasks that do not directly contribute to career advancement [45]) and other unique challenges faced by women |
| Bias-free recommendations | Avoid gender bias in reference and recommendation writing |
| Holistic mentorship | Engage in a relationship encompassing the professional and personal realms |
| Reciprocal relationship | Approach mentoring as a two-way, constructive relationship |
| Transparency and guidance | Provide clear guidance on negotiating pay, benefits, startup packages, e.g., sharing own hire letter |
| Professional development | Share opportunities for professional development and skill building. Consider leaning out from an opportunity and offering it to a mentee. |
| Preparation for reviews | Help mentees with annual evaluations, promotion reviews, job talks by reviewing CV and conducting mock interviews |
| Encouragement of voice | Recognize there are behavioral differences in communications between men and women, pushing women to speak up and project. Deborah Tannen's You Just Don't Understand: Women and Men in Conversation highlight these differences [46]. Permission to Speak focuses on how individuals can use their voice more effectively. Support self-promotion and public engagement by recommending articles such as "Crafting your scientist brand" and books like The New Academic: How to write, present and profile your amazing research to the world [47,48] |
| Belief in potential | Instill confidence in mentees and address self-doubt |
| Continuous support | Provide consistent feedback, advice, and a listening ear |
| Priority setting and career planning | Assist with priorities, career moves, and creating a professional roadmap |
| Inclusive collaboration | Encourage male mentees to support and collaborate with women |

## Public advocacy and actions

**Leaning out and sponsorship.** Participants spoke about men in senior positions who hold on to their positions tightly. Instead, they should support their junior colleagues by stepping down and elevating women in their place.

"...I would say older white men in particular, have serious issues with stepping down. They're so used to power for decades... And so that's why they hang on for a long, long time. Well after their time is up." -P1, M, 70–74

Participants suggested a number of ways that men can push women forward while taking a step back, including: seeking women as collaborators on grants and as first or last authors on papers, offering women opportunities to speak on panels, nominating them for leadership positions and awards, creating positions for women or elevating them as co-leaders, giving them airtime, explicitly acknowledging their leadership and skills, and supporting their work and their journey as leaders. Respondents commented that men should have gender equity as a personal criterion when nominating others while also thinking of representation from low- and middle-income countries.

"I did not want to be one of those people who didn't get out of the way. And the junior folks that I had mentored...They were ready to lead, and as long as I was there, there was never going to be that possibility" -P8, M, 65–69

Another respondent also spoke about the importance of junior men being inclusive and collaborative with women:

"And the more we can talk about collective responsibility and saying that it's important to have diverse teams… even at that early stages of career progression, men are stepping over women and they may not realize it…whether it is to publish first, whether it is to lead a working group, even the closed door invitations to be invited to a particular meeting or discussion, thinking about how young men can make sure to bring another young woman or another underrepresented person with them into all of these kind of opportunities." -P7, F, 40–44

### Role modeling and accommodating caregiving

According to some respondents, men need to be aware of how they are extending offers to socialize after working hours. They should consider the activities they suggest that may prohibit women from participating due to caregiving responsibilities. Some participants pointed out that men can also step up their caregiving duties by modeling this at work. They can make it known when they are leaving early to pick up their children or taking time off to care for their sick child or aging parent. This also applies to advising mentees and employees on work-life integration.

"When she had her first child, she decided to go 80%... She said, '... I want to come home to be with the kids'. And I said, '...So do all the men that I know, and you don't just cut your pay 20%...because you feel guilty that if you do go home in the middle of the day or you go to a sports event, a soccer game… there's not a man I've ever met that feels that way.'" -P1, M, 70–74

Men should know at what stage their female mentees/ employees are at in their lives and recognize the impact of caregiving on their work. Men can support women by allowing them to prioritize their families, giving time and grace periods for tasks and projects, especially during difficult periods or crises. Caregiving can also be a facilitator to connect with employees, students, and mentees.

"I just made some rules for my whole team, and the first rule is family first…children and pets are welcome on all of our virtual platforms… Equity is saying, we're going to help everybody succeed. And the people who are in the middle of childbearing, we get that, we care, and we are going to be supportive." –P8, M, 65–69

### Bystander interventions

Participants described actions by allies to call out sexism, inequitable treatment and behavior, and microaggressions. Those actions included redirecting the conversation or question, speaking up for other women, intervening either one on one or in a group, pointing out inappropriate communication or actions by men about women, and deflecting where appropriate. Additionally, participants spoke about the need for meeting facilitators and panel moderators to call out men for their sexist behavior in the moment or follow up privately afterward. Some participants mentioned that there are certain facilitators to intervening on microaggressions such as intersectionality and not being part of the dominant group. The last sentiment was expressed in the following comment:

"But if it's somebody that's being arrogant and really inappropriate and you don't want this to be perceived as accepted, and you do have to step in the moment…I've stepped in right in the moment and …I've gone afterwards and said, 'She's a doctor too. So, if you're going to do doctor for everyone, you better do doctor for her as well.'" -P15, M, 55–59

"...I was sitting with a junior researcher who was a woman and had two bosses who were male...sitting around her kind of joking about...what a woman's sexual role is. And being able to say, 'well, in my work you wouldn't be able to talk about it like this. And I'm sure it makes her feel very uncomfortable.'" -P9, M, 50–54

### Institutional/Organizational

Some respondents indicated that rather than relying on individual actions, institutions should take the lead in enacting change to ensure gender equity in the workplace. These actions can include creating a hospitable workplace environment and supportive policies, developing goals and metrics, and requiring regular training and evaluations.

"But generally, I think it to be hard for men to make room for women without some institutional push, institutional rewards...So I think the institutions that have created this workspace have more responsibility than the men themselves." -P10, F, 45–49

### Workplace environment

**Workplace culture.** All respondents spoke about workplace culture and the impact on researchers and their career advancement, with several emphasizing that the impacts and solutions are not exclusive to women. Some respondents acknowledged that academic environments breed competition rather than collaboration:

"I think what prevents men is just they're focused on their own career trajectories and just trying to succeed and survive in academia themselves....it is a struggle at our one institution to keep one self-funded and keep productive enough to get those promotion hoops behind you...it's hard when you're in the thick of that to think about making space for others." -P6, F, 50–54

To address the competitive environment, fostering a hospitable workplace culture needs strong leadership who will recognize disparities, advocate for equity, act on any instances of workplace harassment and aggression, and set up checks and balances. Institutions should actively recruit leaders who value diversity and gender equity. Several participants suggested a reframing from a zero-sum game to a rising tide will lift all boats mentality, providing concrete recommendations that institutions could accomplish this by promoting allyship, collective benefits, and engagement by men with women. Institutions can also create spaces for success, promoting a collaborative environment, and a work culture that discourages competition and values everyone's contributions. According to one participant:

"For NIH dollars, there's only so much to go around...And if you do succeed and get tenure… it's easy to be less competitive because you have the protections…It's really on institutions to structure it such that…you should try and ensure that they all succeed. And if everybody knows that, that it's a level playing field and it's not, some kind of hunger games struggle, it's better for everyone." -P8, M, 65–69

Additionally, some respondents spoke about the impact of having a collaborative leadership model in reducing competition and supporting women's advancement where a leadership position is shared between two individuals with different responsibilities.

"I don't know why we had to have a single director… Maybe you have to divide up certain responsibilities by skills if you have to, but …it's just more fun and it's more effective and it's more sustainable because nobody's doing their leadership role as their only job… Institutions should strive to create a legacy of promoting women to senior leadership positions and co-chairing commissions by integrating these actions into their policies and practices." -P5, F, 45–49

### Discussion spaces for men

Several respondents recommended providing men with opportunities to actively and regularly engage in the gender equity conversation. Participants talked about the need for dedicated safe discussion spaces specifically for men outside of

women's conferences and that these should be skillfully structured and facilitated conversations. The spaces should allow for men to voice concerns, ask questions, and exchange knowledge. Men should be able to discuss gender equity, commitment, and accountability. Those spaces could include learning about the differences in communication styles between men and women and addressing related issues. Such spaces and dedicated time could facilitate discussion using case studies and working towards action.

> "...I feel like there will be a lot of men who might be willing to lean out or to be allies, if they have a safe space to have these conversations with women without it being accusatory or pointing their fingers or anything like that…it needs to be a safe space for them to admit their ignorance, but if they don't feel like it's a safe space, they may not, even if inside they want to." -P12, F, 45–49

## Creating workplace peer-to-peer solidarity

Several respondents emphasized the importance of peer-to-peer support to uplift women and reduce competition. Some respondents pointed to the mutual benefits of fostering interpersonal relationships between men and women. Respondents provided recommendations for men to support women colleagues. For some participants, these relationships may arise informally, but institutions could facilitate this through the creation of mixed gender peer mentorship groups or project groups where individuals alternate between taking the lead on a project and providing support. Some recommendations by respondents on actions by male peers include:

- Attending women's talks, giving encouragement and providing feedback at the conclusion
- Reminding women to take credit and acknowledging women publicly for their contributions
- Collaborating on grants
- Transferring legacy/projects to a woman when leaving/retiring from an institution
- Creating career advancement opportunities for women
- Providing peer mentorship
- Sharing successful approaches for public recognition
- Discussing how to navigate career challenges
- Recognizing when a female colleague is overwhelmed or handling competing priorities and offering support

  One respondent spoke about the valuable support she received from male peers:

  "I think about allyship really specifically in terms of … just maybe giving up some of what they have to move things forward... So I have to say that honestly it's peer male colleagues similar generationally, not by age, but generations in terms of when we came in together, what kind of work we've been doing together when we've been on the same teams that have really helped." -P5, F, 45–49

## Meetings, conference panels, commissions, committees

Some respondents provided tools and recommendations for institutions and groups to ensure equitable representation in meetings and conferences. To support equal representation for women, institutions can incorporate a check for gender balance into standard operating procedures when developing invitation lists to high-profile meetings and make space in agendas for women's perspectives, including providing speaking opportunities for underrepresented women.

These suggestions were illustrated by this quote:

"...let's say we are organizing a new conference…a new commission or a panel… right at the start, you've got to say, this conference will be co-chaired by a male and a female. This panel will be co-moderated. And then at the end of this initial round of selection, have we made sure (of)... at least gender parity? …Sometimes it needs to be an all women commission because that's what the topic requires." -P2, M, 50–54

## Policies

Participants discussed the responsibility of institutions to recognize gender disparities, including biological differences that impact women, and enact cultural change by developing policies and guidelines that work towards equity. Policies are necessary to support the sexual and reproductive health and rights of women in the workplace throughout the lifecourse, from menstruation, family planning, childbearing, and menopause, while also protecting them from harassment and violence.

Participants provided several examples of improvements needed in institutional policies (Fig 2).

One participant gave an example of a structural change that he advocated for:

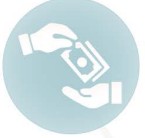
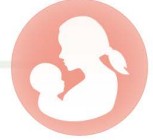
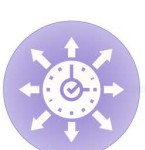
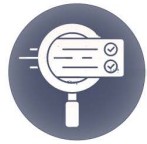
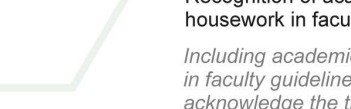
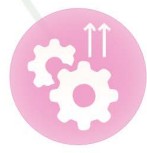

**Transparency in pay and start-up packages**
*Better communication on salaries and benefits to ensure fairness and attract diverse candidates*

**Family-friendly policies**
*Allowing pauses in PhD programs, grants, or tenure tracks for parental leave, providing lactation rooms and mental health support to create a supportive environment for parents and caregivers*

**Offering flexibility in work types and time commitments**
*Providing flexible working arrangements to accommodate different needs and preferences, fostering work-life integration and productivity*

**Changes to promotional packages and annual evaluations**
*Updating promotional criteria to incorporate peer evaluations, equity, mentoring, and community impact, ensuring a comprehensive review of faculty contributions*

**Enhanced representation through supportive services**
*Increasing representation of women through mentorship programs and interactions with female role models to support recruitment and retention*

**Recognition of academic housework in faculty guidelines**
*Including academic housework in faculty guidelines to acknowledge the time and effort spent on administrative and supportive tasks that facilitate the institution's operations*

**Fig 2. Recommendations for institutional policies to promote gender equity.**

"There's some things in the department I'm just cringing…not allowing women to work part-time. That's just ridiculous. Many of them obviously have children...they have these rules in place that they won't take part-time faculty. When I was chief of medicine…we pushed for letting women work and having flexible work hours and having a covering system for each other" -P21, M, 60–64

Another respondent suggested that the way to stimulate institutions to promote gender equity in leadership positions was through funder policies:

"If you were going to capture a group that was going to be able to incentivize more gender equity related to leadership, really it's the funders who could be asked to be looking at what's the proportion of leadership from different genders.… Otherwise, you're just depending on the institution, so of their own goodwill decide that this is a priority, which would be nice." -P16, F, 45–49

### Goals and metrics

Some respondents suggested developing organizational goals to achieve gender equity and then creating metrics on gender parity and salary gaps for accountability and tracking of progress. Alternatively, institutions can devise a scorecard with red, yellow, and green to measure performance and would need to regularly re-examine and revise it. These goals and metrics, when paired with interventions and communications, are an effective tool to monitor gender equity and adapt course if needed. They should inform hiring, promotions, and annual reviews. One respondent suggested a town hall to review metrics and processes that are in place to allow for greater transparency, accountability, and improvement.

**Professional development and evaluation.** Respondents discussed the need for required training and self-assessments to raise awareness of and educate faculty on gender disparities and bias, promote equity, and foster male allyship. Some respondents recommended these topics to be part of leadership training and use positive and inclusive language to engage men effectively. Areas of focus could include understanding biological differences and potential work accommodations, cultural expectations, structural barriers faced by women, teamwork, and mentorship. To encourage allyship, the training could provide a toolkit for men that might include checklists and real-life case studies.

Several respondents recommended that a commitment of allyship be built into promotion packages, salary reviews, or annual evaluations. To gain a full understanding of an individual's allyship these evaluations could include feedback from peers and mentees with recommendations on how to help faculty improve their practices. Based on this feedback, institutions could then recommend trainings and professional development to encourage allyship. Another spoke of a 'citizenship credit' which are extra points faculty can get for non-monetary work (e.g., attending a workshop on gender equity).

### Societal

Participants spoke about changing cultural and societal norms by fostering male allyship at an early age. This could be introduced into formal education at the primary and secondary school levels, with discussions on power, privilege, gender, intersectionality, and gender stereotypes and biases. Some respondents pointed to the importance of families also discussing caregiving, pay equity, and gender norms. Several respondents mentioned the support from male life partners who offered advice, shared responsibilities, and advocated for them.

"I again emphasize more and more that we can [*introduce*] the discrepancies of power as early as we can in education, and if we can get it into primary and secondary education as part of formative education, and then whenever formal education starts for those that enter university and all global health programs should be having a mandatory course on power and privilege and using intersectionality and gender...I think we're going to keep running into those that don't see how they are the enablers, the bystanders, or actually those that are propagating our power imbalances. -P7, F, 40–44

Like the recommendation for academic institutions to offer safe spaces for men to discuss gender, schools and community organizations might organize boys and men's groups where case studies or examples from girls and women could be presented and discussed. According to one respondent:

"But I hear time and time again, even from our elders in indigenous contexts, that… we don't have enough focus on men and men are losing their way, and it perpetuates these kinds of toxic behaviors … And without giving them spaces to help them find their way back, it's going to be really hard to have them be good participants of community and recreate the matriarchal societies that we had prior to colonization because of the inputs from colonial societies on these patriarchal norms." -P20, F, 40–44

Another participant spoke about how ingrained it is in our culture for women to do the caregiving and the difficulties of juggling these responsibilities with a career in global health:

"I pay attention more on how life is affecting women differently than men, especially in global health settings where women have that dual responsibility of showing up as professionals at work, but they're also like their primary caretakers at home and in charge of their kids and their families… Women definitely have disproportionate roles outside of the work environment." -P12, F, 45–49

## Discussion

The purpose of this study was to understand the experiences and perspectives of global health leaders on male allyship and present actions and best practices to support the advancement of women leaders in global health. Not all recommendations from the participants fit neatly into one level, whether individual, institutional, or societal (Fig 1 shows the interconnectedness of the different levels). For example, mentoring is an area where traction can be made at the individual level with more training while also being scaffolded by institutional policies and metrics to support elevating women into visible leadership roles. Women face unique challenges related to biological differences as well as societal and cultural expectations, especially around caregiving.

Both institutions and society will benefit from actions that males take to create an academic environment that discourages a zero-sum game mentality, elevates women to leadership positions, and reduces gender disparities. Changing societal norms about shared caregiving to emphasize the benefits of men as caregivers can lead to healthier families and relationships [50–52]. Greater gender equity can also bolster economies and innovation, leading to shared prosperity [53]. National policy changes such as setting quotas for female political representatives has increased legislative seats for women in Mexico, Nicaragua, and Rwanda, while establishing board representation mandates has resulted in more women's board seats in France and Norway [54].

Our research aligns with findings that the degree to which individuals identify as male allies can be influenced by intersectionality. Participants are more likely to empathize with the challenges women face in academia if they have experienced similar challenges as a member of an underrepresented group. True allyship is characterized by a bidirectional power dynamic with a reciprocal and egalitarian relationship and ongoing learning [38]. Although allies have greater social and organizational power, they do not need to be part of the dominant group, as initially defined in the methods, if they are challenging existing power asymmetries and coupling this with action.

Prior research has shown that marginalized groups find solidarity and support through shared experiences [55]. Peer-to-peer support and collaboration were postulated as an effective allyship method that can improve work environments and career pathways for all genders [56–58]. This needs to happen in tandem with a shift in institutional policies and rewards systems in academia. One study argues for placing a greater emphasis on group contributions rather than just individual [59]. Participants spoke about allyship as a continuous practice

of shifting attitudes and actions that works towards uplifting women and paving the way for them. Recommendations offered by participants for better allyship (including mentorship) have been echoed by research reviews and insights across various academic disciplines and fields [21,60–63]. Another publication offers ten strategies for male allyship to support women in academic medicine that include individual, institutional, communication, and inclusion-based approaches [64]. S1 Table has selected resources and tools offered by participants and the research team to support male allies.

## Limitations

Our findings may not be generalizable to early-career leaders, non-academic settings, or global contexts. The study was restricted to mid and late career male and female global health leaders who were based at academic institutions in the United States and Canada due to funding constraints. Additionally, based on the team's positionality in the U.S., the researchers concluded that local investigators would be better placed to lead further research in other geographies. Although the team selected participants who represented a diversity of perspectives, they did not interview participants who represented some ethnicities, including those who identified as Middle Eastern, Southeast Asian, East Asian, and Latin American, nonbinary, transgender, or individuals with disabilities. The team was also unable to interview any participants based at a historically black college or university. A broader representation of individuals may have yielded richer and more diverse perspectives. For several interviews, the participants had limited time, which prevented the researchers from further probing.

## Next steps

Further research is needed to corroborate the findings of this study in other geographies and investigate the relationship between intersectionality and gender inequities in global health leadership. The team plans to collaborate with local researchers to expand engagement across various regions by gathering feedback from a global survey and a focus group with mid-career African physicians on their recommendations to adapt and disseminate the resources, tools, and recommendations. At the same time, there is a need for public education and advocacy through media engagement and coalition building, and reform of institutional policies and practices to implement these recommendations effectively. While this study focused on male allyship, achieving gender parity will also require allyship by women and prioritizing the elevation of diverse leaders from underrepresented groups, such as transgender and indigenous individuals, people with disabilities, and nationals of low- and middle-income countries, especially as the field seeks to establish more equitable global health partnerships.

## Supporting information

**S1 Text. Interview questions.**
(DOCX)

**S1 Table. Resources and tools to support allyship.**
(DOCX)

## Acknowledgments

The authors extend their gratitude to the research participants for sharing their experiences and insights. They also acknowledge the valuable contributions of Dr. Nicholas Zehner, Dr. Madhukar Pai, Rachel Knopf Shey, Dr. Magali Fassiotto, Dr. Mollyann Brodie, and Dr. Lisa Rogo-Gupta, who provided feedback on the qualitative instrument and offered important resources.

## Author contributions

**Conceptualization:** Amanda Marr Chung, Ola Alani, Michele Barry.

**Formal analysis:** Amanda Marr Chung, Ola Alani.

**Funding acquisition:** Amanda Marr Chung, Michele Barry.

**Investigation:** Amanda Marr Chung, Ola Alani.

**Methodology:** Amanda Marr Chung, Ola Alani, Michele Barry.

**Project administration:** Ola Alani.

**Supervision:** Amanda Marr Chung.

**Writing – original draft:** Amanda Marr Chung, Ola Alani.

**Writing – review & editing:** Amanda Marr Chung, Ola Alani, Michele Barry.

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
