## [Decision Letter · Decision Letter 0]

19 Nov 2025

PGPH-D-25-02685

Male Allyship to Advance Women's Global Health Leadership in the Academy

Dear Dr. Chung,

Thank you for submitting your manuscript to PLOS Global Public Health. After careful consideration, we feel that it has merit but does not fully meet PLOS Global Public Health’s publication criteria as it currently stands. Therefore, we invite you to submit a revised version of the manuscript that addresses the points raised during the review process.

This is an important article and, in my view, a strong contribution to the growing body of knowledge on male allyship in global health. The manuscript has been reviewed by two experts in the field, both of whom raised thoughtful concerns that I encourage the authors to address in their revisions.

From my perspective, the study offers potentially valuable insights, but several of them could be drawn out more clearly. Although you engage with concepts such as allyship and intersectionality, these ideas are currently discussed in a relatively narrow way. I recommend strengthening the conceptual grounding by engaging more deeply with existing literature on these concepts and offering a brief but clear conceptualisation early on. Doing so would help orient readers and better signal what to expect in the findings.

Additionally, while you acknowledge that your sample does not include participants from several ethnic and gender identities, such as Middle Eastern, Southeast Asian, East Asian, Latin American, nonbinary, or transgender individuals, you should also expand on what perspectives these groups might have added. Reflecting on how their inclusion could have enriched your analysis and noting the need for further inquiry to understand how intersectionality operates in global health more broadly would enhance the transparency and reflexivity of the study. Some literature about inequalities faced by people with intersectional identities and those in LIMCs could strengthen your manuscript and encourage research in this field.

We look forward to receiving your revised manuscript.

Kind regards,

Naveed Noor, PhD

Academic Editor

Journal Requirements:

1. Please send a completed 'Competing Interests' statement, including any COIs declared by your co-authors. If you have no competing interests to declare, please state "The authors have declared that no competing interests exist". Otherwise please declare all competing interests beginning with the statement "I have read the journal's policy and the authors of this manuscript have the following competing interests:"

i. Please clarify all sources of financial support for your study. List the grants, grant numbers, and organizations that funded your study, including funding received from your institution. Please note that suppliers of material support, including research materials, should be recognized in the Acknowledgements section rather than in the Financial Disclosure.

ii. State the initials, alongside each funding source, of each author to receive each grant. For example: "This work was supported by the National Institutes of Health (####### to AM; ###### to CJ) and the National Science Foundation (###### to AM)."

iii. State what role the funders took in the study. If the funders had no role in your study, please state: “The funders had no role in study design, data collection and analysis, decision to publish, or preparation of the manuscript.”

iv. If any authors received a salary from any of your funders, please state which authors and which funders.

3. Please ensure that your Ethics Statement is available in its entirety at the beginning of your Methods section, under a subheading 'Ethics Statement'.

4. In the online submission form, you indicated that “The data set is available upon request from the corresponding author.”

3. Uploaded as supplementary information.

5. Some material included in your submission may be copyrighted. According to PLOS’s copyright policy, authors who use figures or other material (e.g., graphics, clipart, maps) from another author or copyright holder must demonstrate or obtain permission to publish this material under the Creative Commons Attribution 4.0 International (CC BY 4.0) License used by PLOS journals. Please closely review the details of PLOS’s copyright requirements here: PLOS Licenses and Copyright. If you need to request permissions from a copyright holder, you may use PLOS's Copyright Content Permission form.

Potential Copyright Issues:

Figures 1 and 2: Please confirm whether you drew the images / clip-art within the figure panels by hand. If you did not draw the images, please provide (a) a link to the source of the images or icons and their license / terms of use; or (b) written permission from the copyright holder to publish the images or icons under our CC-BY 4.0 license. Alternatively, you may replace the images with open source alternatives. See these open source resources you may use to replace images / clip-art:

- https://openclipart.org/

Reviewers' comments:

Reviewer's Responses to Questions

**Comments to the Author**

1. Does this manuscript meet PLOS Global Public Health’s publication criteria? Is the manuscript technically sound, and do the data support the conclusions? The manuscript must describe methodologically and ethically rigorous research with conclusions that are appropriately drawn based on the data presented.? Is the manuscript technically sound, and do the data support the conclusions? The manuscript must describe methodologically and ethically rigorous research with conclusions that are appropriately drawn based on the data presented.

Reviewer #1: Yes

Reviewer #2: Yes

2. Has the statistical analysis been performed appropriately and rigorously?

Reviewer #1: N/A

Reviewer #2: N/A

3. Have the authors made all data underlying the findings in their manuscript fully available (please refer to the Data Availability Statement at the start of the manuscript PDF file)?

The PLOS Data policy requires authors to make all data underlying the findings described in their manuscript fully available without restriction, with rare exception. The data should be provided as part of the manuscript or its supporting information, or deposited to a public repository. For example, in addition to summary statistics, the data points behind means, medians and variance measures should be available. If there are restrictions on publicly sharing data—e.g. participant privacy or use of data from a third party—those must be specified.requires authors to make all data underlying the findings described in their manuscript fully available without restriction, with rare exception. The data should be provided as part of the manuscript or its supporting information, or deposited to a public repository. For example, in addition to summary statistics, the data points behind means, medians and variance measures should be available. If there are restrictions on publicly sharing data—e.g. participant privacy or use of data from a third party—those must be specified.

Reviewer #1: No

Reviewer #2: No

4. Is the manuscript presented in an intelligible fashion and written in standard English?

Reviewer #1: Yes

Reviewer #2: Yes

Reviewer #1: Manuscript Number: PGPH-D-25-02685 Review Report Male Allyship

Overall summary of the review

Strength

This manuscript addresses the important topic of male allyship in advancing women’s leadership in global health academia, using a well-structured qualitative approach and presenting findings across individual, institutional, and societal levels. Strengths include clear objectives, rigorous methods, and practical insights for policy and leadership.

Areas that need improvement include:

• Clarify the study type in the title and make it action-oriented.

• Expand geographic representation beyond the U.S. and Canada.

• Provide context for participant quotes.

• Report data saturation and response rates.

• Improve readability by breaking up long sentences and structuring results around clear subthemes.

• acknowledge limitation for geographic representation beyond the U.S. and Canada

Point by point feedback

Title: - Male Allyship to Advance Women's Global Health Leadership in the Academy

Strength

• It addresses a timely and high-Impact Topic: The study area is forgotten by global community especially in academia on the importance of gender equity and leadership.

• The title accurately reflects the manuscript’s core theme male allyship in academic global health leadership.

• Timely and High-Impact Topic: The study addresses a significant gap in global health leadership literature—moving beyond simply identifying barriers to focusing on actionable solutions (allyship and sponsorship). This shift in focus is highly valuable for policymakers and institutional leaders

Weakness

• The short and long title of the manuscript is the same; no difference

• The title talks about two things one “Global Health Leadership” and second “the academic institution in North America”. the focus of study population is not clear

• The kind of the study type is no clearly indicated in the title.

• The phrase “in the Academy” could be slightly ambiguous to international audiences—consider specifying “in academic global health institutions.”

Suggested Revision

• It is better if the title is an action oriented Like “Exploring Male Allyship to Advance Women’s Leadership in Global Health Academia: A Qualitative Study” or Male Allyship to Advance Women’s Leadership in Global Health Academia: A Qualitative Study

Abstract

Strengths:

Introduction

• Well-structured abstract with clear objectives, methods, and findings.

• Strong justification for the study, citing global gender disproportionate disparities and responsibilities gaps in global health leadership.

• Appropriate Methodology: It used a qualitative, semi structured interview approach to explore the perception, experiences and perceptions of high level leaders. Clear thematic analysis

• It use clear research question on the experience of global leaders

• Findings: Clearly structured around three levels—individual male ally, institutional, and societal level which offers a useful conceptual framework to shift cultural norms on gender roles for practical implications

Area of improvement:

• Introduction looks like an advocacy; no clearer separation between background and rationale.

• • Participants were drawn only from the U.S. and Canada, which limits the geographic coverage of the study. In addition, selecting participants exclusively from the WomenLift Health network may reduce the diversity of perspectives and potentially affect the credibility of the findings.

• Conceptual clarity: The term male allyship could be briefly defined in one sentence for clarity.

• The conclusion partially repeats ideas from the introduction it looks like the summary of the introduction and better if based on the findings and highlighting the practical or policy relevance to enhance the impact

Main Manuscript

• The study team approached participants via email and invited them to participate and conduct the interview via zoom; so why the study participants restricted to two high-income countries?

Result

• Line 178-183: The results section reports the number of participants but does not indicate whether data saturation was achieved. In qualitative research, explaining when and how saturation was reached strengthens the credibility and adequacy of the sample size.

• Reporting the response rate as a percentage would help readers interpret the level of participation more easily.

• Line 185-191: The paragraph communicates the general findings, better if key themes are mentioned to give readers a concrete examples

• Line 202: The quote is powerful, but providing brief context about the respondent role, experience, or institution better if included.

• Line Motivation and incentives it is more plausible to read if concrete example of participant quotes are included

• Line 252 to 278: the section contains valuable insights, but some sentences are long and dense. Breaking them into shorter sentences will improve readability. Consider structuring the section around clear subthemes: like awareness, role modeling etc.,

• Line 332-341 quote -P7, F, 40-44 is clear and conventional; however it’s a bit long and better if lightly edit for readability while retaining authenticity

Discussion

Strength

• It is strong in structure, logic, and scholarly tone. It clearly links findings to prior research and offers practical and theoretical insights

• Line 633 Toking et al reference number (54). Line 638 Sinha et al (59) need consistency

Limitation

Strength: Clearly states the population studied & acknowledges selection of diverse perspectives

Area of Improvement: Findings may not generalize to early-career leaders, non-academic settings, or global contexts.

Reviewer #2: This paper examines an important topic: male allyship in global health. A key strength of the paper is its focus on solutions as reported by participants. This helps move the focus away from challenges women in global health face, to exploring actual solutions to the problem. The paper presents many examples of what male allyship looks like, many of these being very actionable. I commend the authors for this. That said, there are several areas that need attention.

1. The findings could be synthesized and made more concise. The authors present A LOT of information, which is overwhelming. There are many ways to address this. First, the authors can use much shorter direct excerpts from participants. Second, the authors could move participant excerpts into a table. Third, the authors could synthesize their findings much more which would reduce the number of themes/issues and tell the story a bit differently. The paper would read better if they structured the results based on the subheadings used in Fig 1. It would help tell a more succinct story.

2. It is not clear what Fig 2 is showing.

3. I wonder if to would help to present the findings by gender, that is, by showing similarities and differences between female and male participants.

4. Could the authors present more details on the study participants. Sone of the content in the excerpts suggests that they were from academic institutions. A demographics table would help, in terms of academic vs non-academic institutions etc. The authors should also provide more details on the sampling and recruitment method: how exactly did they find these 21 participants?

5. The Discussion could be strengthened by a more critical analysis of the concept of male allyship. The authors should consider what their findings/study implications are for the potential and pitfalls of the concept.

**Do you want your identity to be public for this peer review?** For information about this choice, including consent withdrawal, please see our Privacy Policy..

Reviewer #1: No

Reviewer #2: **Yes:**Tsitsi B MasvawureTsitsi B MasvawureTsitsi B MasvawureTsitsi B Masvawure

---

## [Editor Report · Decision Letter 1]

11 Mar 2026

Male Allyship to Advance Women's Leadership in Global Health Academia: A Qualitative Study

PGPH-D-25-02685R1

Dear Dr. Chung,

We are pleased to inform you that your manuscript 'Male Allyship to Advance Women's Leadership in Global Health Academia: A Qualitative Study' has been provisionally accepted for publication in PLOS Global Public Health.

Best regards,

Naveed Noor, PhD

Academic Editor